# Courtship behaviour reveals temporal regularity is a critical social cue in mouse communication

**Catherine Perrodin[1]\*, Colombine Verzat[1,2], Daniel Bendor[1]\***

[1]Institute of Behavioural Neuroscience, Department of Experimental Psychology, University College London, London, United Kingdom; [2]Idiap Research Institute, Martigny, Switzerland

**Abstract** While animals navigating the real world face a barrage of sensory input, their brains evolved to perceptually compress multidimensional information by selectively extracting the features relevant for survival. Notably, communication signals supporting social interactions in several mammalian species consist of acoustically complex sequences of vocalisations. However, little is known about what information listeners extract from such time-varying sensory streams. Here, we utilise female mice's natural behavioural response to male courtship songs to identify the relevant acoustic dimensions used in their social decisions. We found that females were highly sensitive to disruptions of song temporal regularity and preferentially approached playbacks of intact over rhythmically irregular versions of male songs. In contrast, female behaviour was invariant to manipulations affecting the songs' sequential organisation or the spectro-temporal structure of individual syllables. The results reveal temporal regularity as a key acoustic cue extracted by mammalian listeners from complex vocal sequences during goal-directed social behaviour.

**\*For correspondence:**
catherine.perrodin@alumni.epfl.ch (CP);
d.bendor@ucl.ac.uk (DB)

## eLife assessment

This **valuable** work advances our understanding of the acoustic features driving the attraction of female mice to male vocalisations. The evidence supporting the conclusions is **solid**, with well-designed place preference assays and manipulations of male song structure. The work will be of broad interest to neurobiologists and ethologists working on mouse social interactions, auditory processing and communication.

## Introduction

Animals behaving in the real world need to efficiently discriminate and monitor relevant sensory input, such as information communicated by conspecifics, from a constantly evolving, complex, and multidimensional barrage of stimulation. In many social species, including humans, vocal communication takes the form of time-varying sequences of acoustic elements and demands a rapid, socially appropriate behavioural response. Yet how the listener's brain effectively extracts salient information from acoustically complex vocal patterns during social goal-directed behaviour remains unclear. Here, we asked which acoustic cues from communication sequences were informative to female mouse listeners when responding to male courtship songs.

Mice, like most other mammals, use complex acoustic patterns to communicate with each other in a number of different social contexts (*Portfors, 2007*; *Liu et al., 2003*). In particular, adult males produce rhythmic sequences of discrete, frequency-modulated pure tone elements in response to sensing the recent presence of a fertile female (*Holy and Guo, 2005*; *Chabout et al., 2012*; *Chabout*

*et al., 2017*). These courtship songs are attractive to sexually receptive females (*Pomerantz et al., 1983*), who respond to song playback with approach behaviour (*Asaba et al., 2014b*; *Shepard and Liu, 2011*; *Chabout et al., 2015*; *Musolf et al., 2015*; *Hammerschmidt et al., 2009*; *Asaba et al., 2017*). Male songs facilitate copulatory success (*Nomoto et al., 2018*) and are thought to serve as fitness displays (*Egnor and Seagraves, 2016*; *Hoffmann et al., 2012*).

Female mice are able to detect and use acoustic information in male vocalisations. On the one hand, reinforcement learning experiments have shown that mouse listeners are able to detect and report spectro-temporal differences between individual ultrasonic call elements (*Neilans et al., 2014*; *Holfoth et al., 2014*), indicating that mice are *perceptually* sensitive to a wide range of acoustic features in the vocalisations. On the other hand, experiments using ethologically relevant place preference paradigms without external reward have shown that females use, or are *behaviourally* sensitive to, certain acoustic characteristics in male songs to guide social decisions, such as discriminating between social contexts (*Chabout et al., 2015*), singer species (*Musolf et al., 2015*), strain (*Asaba et al., 2014a*; *Sugimoto et al., 2011*), and kin (*Musolf et al., 2010*). However, which of the many acoustic cues available in male courtship songs are used by motivated female listeners during natural behaviour remains unclear.

In this study, we asked how listeners process and use complex vocal sequences during natural behaviour. We exploited female mice's natural behavioural response to playbacks of male courtship songs in order to independently manipulate and identify which of the several candidate acoustic features are monitored by the listeners. We found that female approach behaviour was highly sensitive to disruptions of song temporal regularity, but that it was not affected by global manipulations of the songs' sequential structure, or the local removal of syllable spectro-temporal dynamics. The results highlight temporal regularity as a key social cue monitored by female listeners during goal-directed social behaviour.

## Results

### Acoustic characteristics of vocal sequences emitted by male C57Bl/6 mice

In order to study female listeners' behavioural response to vocal sequences, we first collected a database of male C57Bl/6 mice vocalisations for acoustic stimulation in playback experiments. Following a previously published protocol (*Chabout et al., 2017*), we obtained audio recordings of ultrasonic vocalisations emitted by males in response to the presentation of mixtures of urine samples collected from conspecific females in oestrus. This paradigm was deemed optimal for generating stimuli to use in ethologically inspired playback experiments as vocalisations emitted in this context are emitted by solitary males and reflect long-range acoustic 'courtship' displays with the purpose of attracting a female (*Nomoto et al., 2018*; *Portfors and Perkel, 2014*; *Matsumoto and Okanoya, 2016*). The vocal patterns emitted in this context consisted of sequences of individual frequency-modulated calls (e.g. see *Figure 1A*, *Supplementary file 1*; *Holy and Guo, 2005*; *Chabout et al., 2015*; *Tschida et al., 2019*; *Hage et al., 2013*; *Castellucci et al., 2018*). Across the set of recorded male vocalisations, the duration of continuous call elements, or syllables, followed a bimodal distribution, with a majority of short syllables (local maximum = 23 ms, 86% [2039/2373] of recorded syllables with durations shorter than 63 ms [local minimum]) and a minority of long syllables (local maximum = 88 ms, 14% [334/2373] of recorded syllables with durations longer than 63 ms, *Figure 1B*). These male calls were restricted to the ultrasonic frequency range, with the maximal energy in individual syllables occurring around 76 kHz (median peak frequency and 95% confidence interval [CI], 76 ± 0.3 kHz, n = 2373 syllables) and spanning a bandwidth of 16 kHz ([68–84 kHz]; *Figure 1C*). Individual syllables were organised in temporally regular sequences and often started at an approximately constant time delay from each other (*Holy and Guo, 2005*; *Castellucci et al., 2018*), as evidenced by the narrow concentration of inter-syllable interval durations around 104 ± 2 ms (median inter-syllable interval and 95% CI; inter-quartile range of syllable interval = 61 ms, n = 2322 syllables followed by another syllable within 2 s; *Figure 1D*). From this set of ultrasonic vocalisations, we selected a smaller, representative sample for use in subsequent playback experiments (n = 957 syllables, dark grey histograms in *Figure 1B–D*). This stimulus set was composed of seven distinct songs produced by C57Bl/6 male mice aged 22 wk on average (median age, range [10–27 wk]) in response to the presentation of female urine. The songs

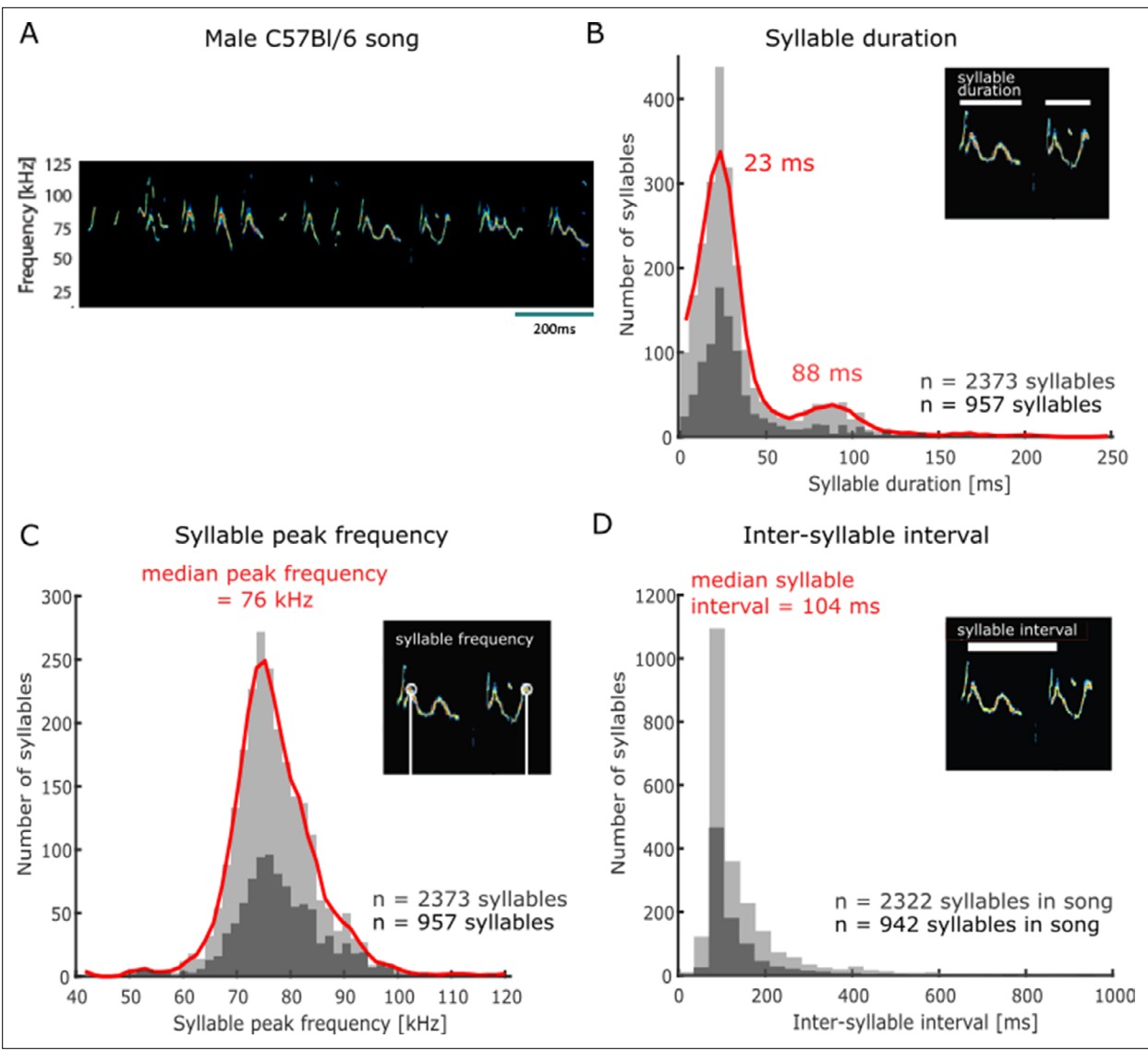

**Figure 1.** Acoustic features of C57Bl/6 mouse courtship songs. (**A**) Spectrogram of a segment of ultrasonic male vocalisations used for stimulation. (**B**) Distribution of individual ultrasonic syllable durations across a large set of male mouse vocalisations (n = 2373 syllables, light grey histogram; red line is smoothed distribution), including the stimulus set used in subsequent playback experiments (n = 957 syllables, dark grey), emitted in response to the presentation of urine samples from females in oestrus. (**C**) Distribution of syllable peak frequency (point of maximum amplitude across the call element in kHz) across all recorded syllables (n = 2373,, light grey), and the subset of syllables used for playback (n = 957, dark grey). (**D**) Distribution of inter-syllable interval durations. The analysis was restricted to syllables with a subsequent syllable starting within 2 s (all recorded syllables: n = 2322, light grey; playback stimulus set: n = 942, dark grey).

varied in duration, lasting on average 33.4 s (median duration, range [13–41 s]), and were composed of an average of 145 syllables (median, range [95–186]).

## Female mice preferentially approach playbacks of male songs over silence

Using our pre-recorded subset of male vocal sequences for stimulation, we then aimed to confirm that female mice displayed a preferential approach response to male songs in our ethologically inspired behavioural paradigm. We used a place preference assay to evaluate female listeners' behaviour in response to the playback of our pre-recorded song set in a two-compartment behavioural chamber (*Figure 2A*). Importantly, olfactory stimulation from mixed male bedding samples placed under each of the loudspeakers was present throughout the testing session in order to increase sexual arousal

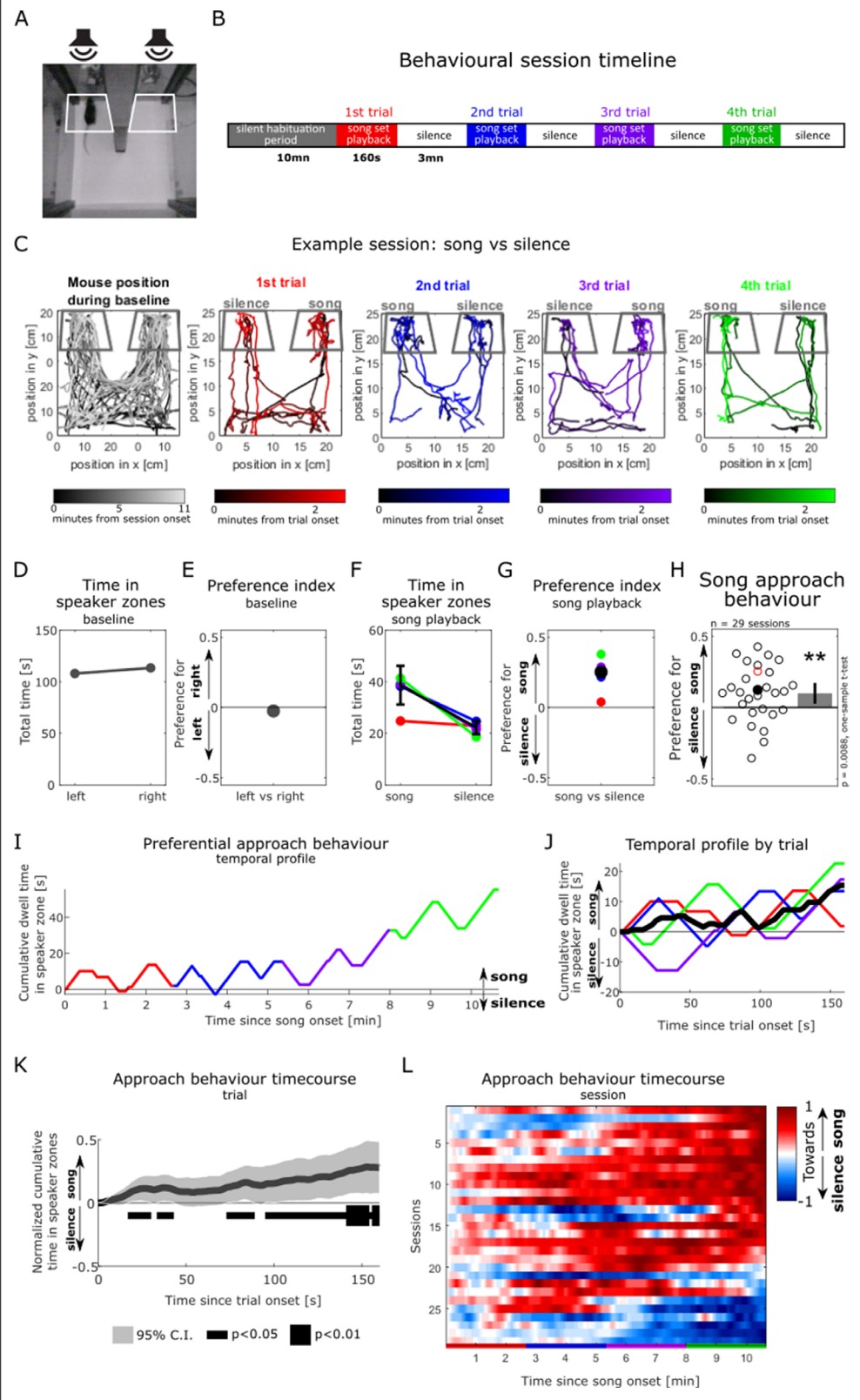

**Figure 2.** Female mice preferentially approach playbacks of male songs over silence. (**A**) Video frame showing the testing box with a soundproof partition (middle) positioned between two ultrasonic loudspeakers. White outlines show the two 'speaker zones' used as regions of interest for quantifying the animal's position. (**B**) Experimental timeline. (**C**) Tracking of the animal's position during an example behavioural session, in which a silent baseline

*Figure 2 continued*

period (leftmost panel) is followed by the playback of intact songs from one side contrasted with silence from the other side (coloured panels). Colour saturation indicates time since start of the experimental period of interest. Dark grey outlines indicate the speaker zones. (**D**) Time the animal spent in the speaker zones during the silent baseline period in (**C**). (**E**) Index quantifying the animal's relative place preference to the left vs right speaker zones during the silent baseline. This index was computed as the difference between the time spent in either speaker zones, normalised by the sum of the time in both speaker zones (see 'Methods'). A preference index value close to zero indicates no preference to either side. (**F**) Time spent in the speaker zones corresponding to song playback and silence during each song presentation trial (coloured circles). The black circle shows the median of the four sound presentation trials. Error bars: standard error of the mean. (**G**) Preference index in response to the playback of intact songs compared to silence over the course of the example session. The black circle shows the median of the four sound presentation trials. The preference index is the difference between the time in the song playback and the silent speaker zones, divided by the sum of the time in both speaker zones. A positive index value reflects the animal's preferential approach to the sources of song playback over silence. (**H**) Population summary of female approach response to playback of intact male songs (positive values) over silence (negative). Each circle is the preference index displayed by individual animals in one behavioural session (median of four sound presentation trials). The red circle corresponds to the example session in (**C**) and (**G**). Open circles: sound playback at 58 dB SPL, filled circles: 68 dB SPL. Bar plot shows mean preference index across sessions and 95% confidence interval (CI). One-sample two-tailed *t*-test, \*\*p<0.01. (**I**) Temporal profile of approach behaviour over the four sound presentation trials in the example session in (**C**), calculated as the cumulative sum of time in the intact song playback (positively weighted) vs silent (negatively weighted) speaker zone. (**J**) Trial-averaged profile of approach behaviour to song playback in the example session, calculated as in (**I**). (**K**) Population-averaged approach behaviour time course in response to intact mouse songs vs silence, calculated as in (**J**). The dark grey trace indicates the mean of trial-based temporal profiles across all sessions (n = 29). For each session, the median of four sound presentation trials (e.g. black trace in **J**) was normalised to its maximal amplitude. The horizontal black bar indicates time bins during the course of a sound playback trial in which the cumulative approach behaviour significantly deviates from zero (one-sample two-tailed *t*-test). (**L**) Normalised temporal profiles of approach behaviour to mouse songs vs silence over the course of four sound presentation trials (x-axis, coloured bars) for each of the behavioural sessions (y-axis, each animal is one line, n = 29), calculated as in (**I**). Sessions (lines) are ordered by the amplitude of their last element. (**F, G, I, J**) Black traces indicate the session average (median) across the four sound presentation trials (coloured traces).

The online version of this article includes the following video and figure supplement(s) for figure 2:

**Figure supplement 1.** No difference in listener age across playback experiments.

**Figure supplement 2.** Approach behaviour is consistent across varying speaker zone lengths.

**Figure 2—video 1.** Example video tracking of mouse approach behaviour.
https://elifesciences.org/articles/86464/figures#fig2video1

**Figure 2—video 2.** Example video tracking of mouse approach behaviour.
https://elifesciences.org/articles/86464/figures#fig2video2

and motivation in the female listeners (*Hammerschmidt et al., 2009*; *Asaba et al., 2014a*) and create a multisensory coherent approximation of a natural social situation. Following a silent habituation period during which the animal was free to explore the behavioural box (at least 10 min; grey traces in *Figure 2B and C*), one-sided playback of male vocalisations was initiated while the second speaker remained inactive (red traces in *Figure 2B and C*). The pre-recorded set of seven songs was repeated four times during one behavioural session, with the order of individual songs randomised in each trial, and the playback side alternating between trials (*Figure 2B and C*). A female mouse's approach response to song playback was measured by quantifying the time the animal spent within a 'speaker zone' over the course of a behavioural session (white [grey] trapezoidal outlines in *Figure 2A and C*, respectively). Young female C57Bl/6 mice (aged 5–11 wk; *Figure 2—figure supplement 1*) participated once in the playback experiment, while in the fertile (proestrus or oestrus) stage of their oestrous cycle (as assessed by vaginal cytology, see 'Methods').

In the example session illustrated in *Figure 2*, the mouse occupied both speaker zones equally during the silent baseline (*Figure 1D*) and the first sound playback trial (red trace in *Figure 2F and I*). Over the course of the three subsequent trials, the mouse displayed a consistent preferential approach towards the source of male song playback over the silent speaker, irrespective of the playback side (blue, purple, and green traces, *Figure 2F and I*, *Figure 2—videos 1 and 2*). An animal's

approach behaviour over the course of an entire session can be quantified using a preference index (see 'Methods' and *Figure 2E and G*), which accurately captures the cumulative preferential dwell time at the song playback vs silent speaker (see correspondence between preference index values in *Figure 2G* and endpoints of trial temporal profiles in *Figure 2J*). This preference index metric was also shown by a number of previous studies to be the most sensitive readout for similar assays of female approach to male song playback (*Musolf et al., 2015*; *Hammerschmidt et al., 2009*; *Asaba et al., 2014a*). In this example, the mouse preferentially approached the song playback over silence in ¾ trials, resulting in a preference index value of 25.3% (median of four trials; black dot in *Figure 2G*). The behaviour of female listeners tested in this paradigm significantly discriminated between song playback and silence (n = 29 sessions, mean preference index significantly different from 0, one-sample *t*-test, $t(28) = 2.81$, p=0.0088; *Figure 2H*), and mouse listeners were overwhelmingly attracted to the playback of our subset of male songs. The preference for song over silence was robust to different lengths of the speaker zone used to calculate the preference index (*Figure 2—figure supplement 2A and B*). Mouse listeners demonstrated significant approach response after as little as 16.9 s of sound playback across the four sound presentation trials (normalised cumulative time in speaker zone significantly different from 0, one-sample *t*-test, p<0.05; *Figure 2K*), with preference further increasing over the course of song playback. The build-up of preferential approach to song was evident in the vast majority of animals, mostly sustained over a behavioural session (*Figure 2L*), and similar across the four sound presentation trials (*Figure 2—figure supplement 2B*). Together, these results replicate previous work by other laboratories (*Asaba et al., 2014b*; *Shepard and Liu, 2011*; *Chabout et al., 2015*; *Musolf et al., 2015*; *Hammerschmidt et al., 2009*) and confirm the feasibility of recreating an ethologically driven behavioural assay of female approach response to male song playback in the controlled laboratory environment.

## Female approach behaviour is not affected by changes to global song structure

Given this proof of principle that socially motivated female listeners approach the source of male song playback, we then extended this natural behavioural paradigm to directly evaluate how listeners perceive and use acoustic features from vocal sequences. Specifically, we tested the hypothesis that, during naturalistic goal-directed behaviour in a social context, female listeners perceptually compress the high sensory dimensionality of male songs by selectively monitoring a reduced subset of meaningful acoustic features in isolation. We quantified females' place preference in the same behavioural assay in response to the near-simultaneous playback of the pre-recorded set of male songs from one speaker and an acoustically modified version of the songs from the second speaker. Given that females preferentially approach the speaker playing the male song, differences in the relative time spent at the source of intact vs manipulated sound playback are interpreted to demonstrate the behavioural relevance of the manipulated acoustic dimension. In contrast, equal approach behaviour to both intact and manipulated song playback is interpreted as females not perceiving, or not relying on, the tested acoustic dimension in their natural behavioural response to courtship song.

In the first instance, we evaluated whether listeners would be sensitive to two types of global manipulations affecting the song structure at longer timescales. Building on previous work suggesting that female mice use syntactic information to discriminate songs produced in different social contexts (*Chabout et al., 2015*), we tested whether randomising the order of the syllables in the song would affect their approach behaviour (*Figure 3A*, left, and *Supplementary file 2*). We found that the sequential organisation of syllables in the male songs was not necessary for normal approach behaviour (mean preference index in response to intact vs randomised songs did not differ from zero, one-sample *t*-test, $t(20) = 0.126$, p=0.90; *Figure 3A*, right). This behavioural invariance to changes in the syllable order was apparent at all timepoints of song set playback (*Figure 3B*), as well as at different speaker zone lengths (*Figure 2—figure supplement 2C*), and was comparable across the four sound presentation trials (*Figure 3—figure supplement 1D*). This confirms that the lack of sensitivity to syllable sequence structure we observe here was not caused by the selection of specific temporal or spatial parameters for analysis.

Next, we contrasted intact songs with a time-reversed version of the songs. While preserving the long-term spectra and range of spectral changes, this manipulation generated sounds with novel temporal features that different subsets of auditory neurons are tuned to, such as reversing the

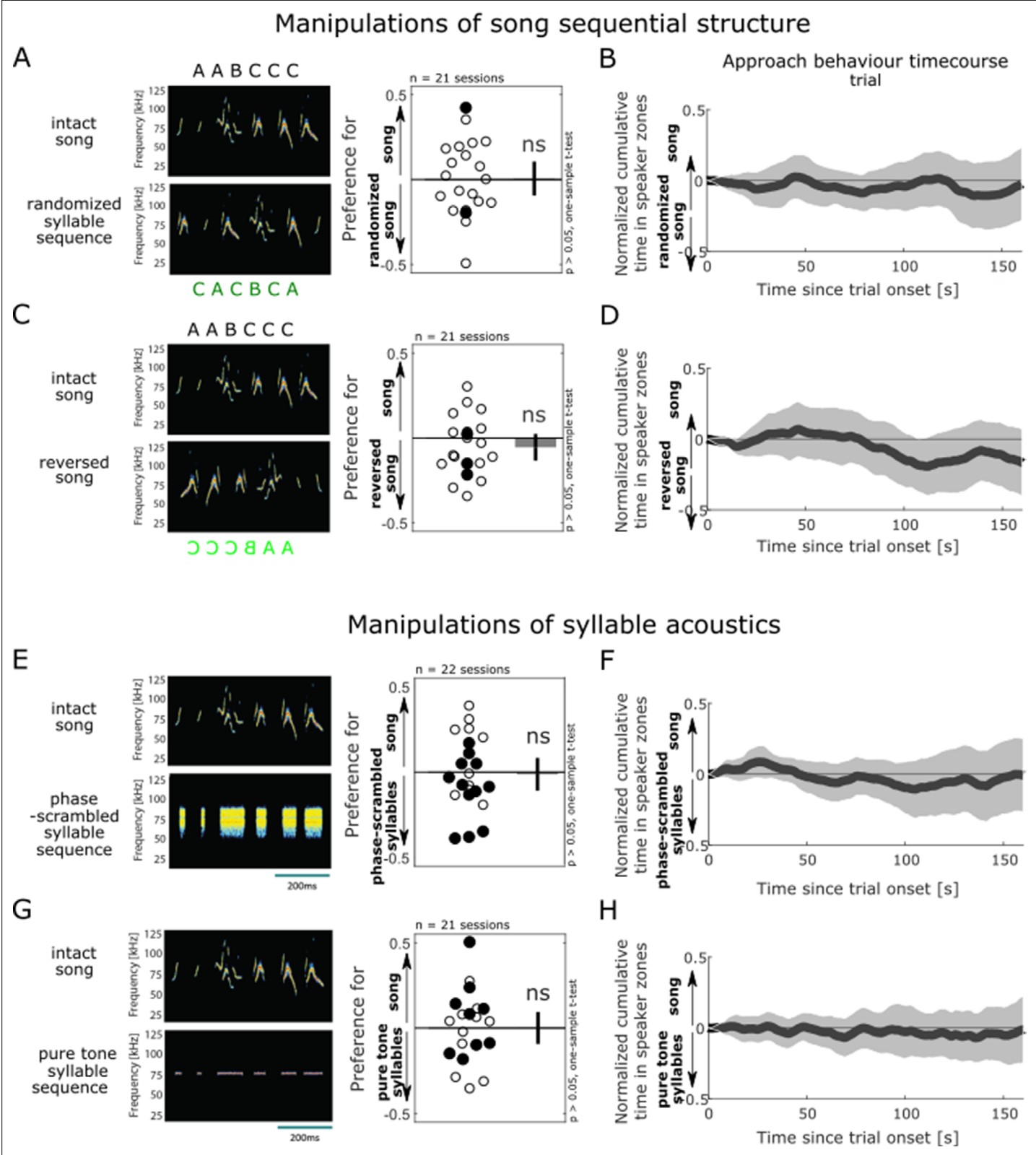

**Figure 3.** Female approach behaviour is not affected by changes to global song structure or the removal of syllable spectro-temporal dynamics. (**A**) Female approach behaviour during simultaneous playback of intact male songs (top) and corresponding randomised syllable sequences (bottom), displayed as in *Figure 2H*. Each circle is the preference index displayed by individual animals in one behavioural session (median of four sound presentation trials). Open circles indicate sound playback at 58 dB SPL, filled circles at 68 dB SPL. Bar plot shows mean preference index across sessions,

*Figure 3 continued on next page*

*Figure 3 continued*

and error bar show 95% confidence interval (CI). One-sample two-tailed *t*-test, ns: non-significant, p>0.05. (**B**) Population time course of approach behaviour during the playback of intact (positively weighted) vs randomised songs (negatively weighted), displayed as in *Figure 2K*. The dark grey trace indicates the population mean of trial-based temporal profiles across all sessions (n = 21 sessions). For each session, the median of four sound presentation trials (e.g. black trace in **J**) was normalised to the absolute value of its maximal amplitude. At no time bins during the course of a sound playback trial did the cumulative approach behaviour significantly deviate from zero (one-sample two-tailed *t*-test, all p>0.05). (**C**) Playback of intact songs (top) contrasted with temporally reversed songs (bottom). (**D**) Typical time course of approach behaviour during the playback of intact (positively weighted) vs reversed songs (negatively weighted). (**E**) Playback of intact songs (top) and sequences of phase-scrambled syllables (bottom). (**F**) Typical time course of approach behaviour during the playback of intact (positively weighted) vs phase-scrambled syllable sequences (negatively weighted). (**G**) Playback of intact songs (top) contrasted with sequences of pure tones (bottom). (**H**) Typical time course of approach behaviour during the playback of intact (positively weighted) vs pure tone sequences (negatively weighted).

The online version of this article includes the following figure supplement(s) for figure 3:

**Figure supplement 1.** Trial-based temporal profiles of approach behaviour.

direction of frequency trajectories (*Issa et al., 2017*; *Lui and Mendelson, 2003*; *Sollini et al., 2018*; *Tian and Rauschecker, 1998*; *Fuzessery, 1994*; *Razak and Fuzessery, 2006*; *Geis and Borst, 2013*; *Figure 3C*, left, and *Supplementary file 3*). Temporal reversal of the vocal sequence did not significantly affect females' approach behaviour (mean preference index in response to intact vs reversed songs did not differ from zero, one-sample *t*-test, *t*(20) = –1.46, p=0.16; *Figure 3C*, right). This lack of behavioural sensitivity to song temporal reversal was present throughout sound playback in the vast majority of animals tested (*Figure 3D*), across the sound presentation trials (*Figure 3—figure supplement 1E*), and for different spatial regions of interest (*Figure 2—figure supplement 2*). Together, these observations suggest that female listeners are not relying on global cues reflecting the overall structure of male song sequences when selecting which sound source to approach.

## Female approach behaviour is robust to the removal of syllable spectro-temporal dynamics

Since auditory neurons are typically tuned to specific spectro-temporal features in the song syllables, and mice are able to use those features to discriminate between individual syllables (*Neilans et al., 2014*; *Holfoth et al., 2014*), we next wondered whether manipulations disrupting the fine acoustic structure of individual calls in the male vocal sequences would be more easily discriminated by female listeners. We first created a phase-scrambled version of the songs that preserved the average frequency spectrum of the syllables, but removed the spectro-temporal dynamics of individual syllables and shaped each noise burst with a flat envelope (*Figure 3E*, left, and *Supplementary file 4*). Females approached the intact and phase-scrambled song playback similarly (mean preference index in response to intact vs phase-scrambled songs did not differ from zero, one-sample *t*-test, *t*(22) = –0.27, p=0.79; *Figure 3E*, right). Second, comparing intact songs with more drastically simplified sound sequences that stripped down each syllable into a pure tone at the peak syllable frequency (76 kHz) also failed to elicit differential approach behaviour (mean preference index in response to intact vs pure tone sequences did not differ from zero, one-sample *t*-test, *t*(20) = 0.034, p=0.97; *Figure 3G* and *Supplementary file 5*). These two instances of behavioural insensitivity to changes in the syllable acoustic trajectories were robust to variations in temporal (*Figure 3F and H*, *Figure 3—figure supplement 1F and G*) and spatial (*Figure 2—figure supplement 2D and G*) parameters used for analysis. Together, these findings show that female behaviour was robust to the removal of fast spectro-temporal structure in the syllables, at least when sound energy was still present at the natural syllable peak frequency.

## Female approach behaviour is reduced by the disruption of song temporal regularity

Lastly, following the earlier observation that syllable onsets occur at consistent time intervals from each other across the song sequences (see *Castellucci et al., 2018* and *Figure 1D*), we directly probed whether female listeners would be sensitive to manipulations of the temporal regularity in the male song. To do this, we generated a set of temporally irregular songs that consisted of the same syllable sequence with a broadened distribution of silent pauses compared to intact songs (interquartile range of syllable interval in irregular songs = 134 ms compared to 61 ms for intact songs; *Figure 4A, B and*

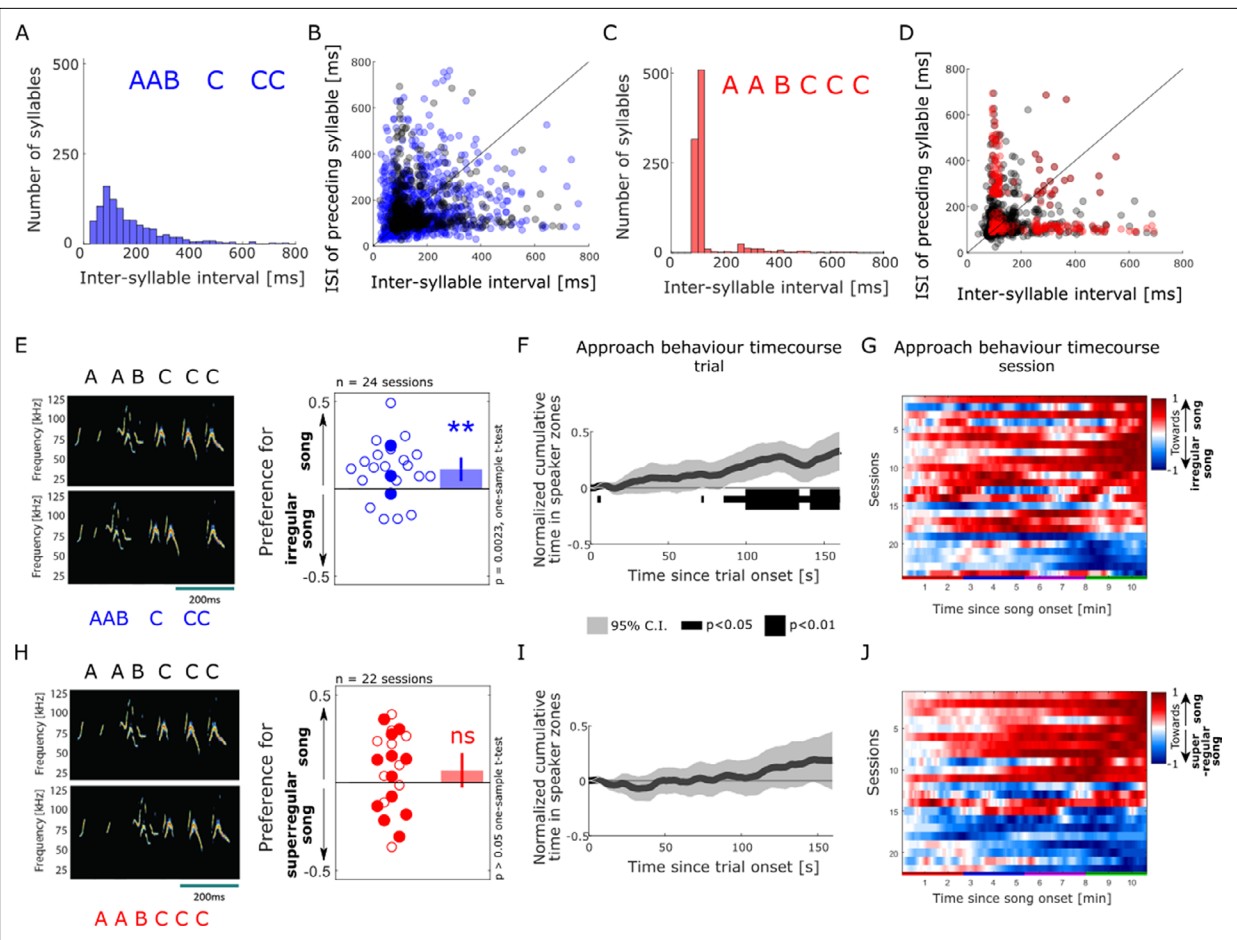

**Figure 4.** Female approach behaviour is sensitive to disruption of courtship temporal regularity. (**A**) Distribution of inter-syllable interval (ISI) durations across the set of temporally irregular songs, calculated as in *Figure 1D*. (**B**) Sequential relationships of ISI durations in the intact (shaded grey dots) and temporally irregular (shaded blue dots). (**C**) Distribution of ISI durations across the super-regular song set. (**D**) Sequential relationship of ISI durations in the intact (shaded grey dots) and super-regular (shaded red dots). (**A–D**) n = 957 syllables. (**E**) Female approach behaviour during simultaneous playback of intact male songs (top) and temporally irregular songs (bottom), displayed as in *Figure 2H*. Each circle indicates the preference index displayed by individual animals in one behavioural session (median of four sound presentation trials). Open circles: sound playback at 58 dB SPL, filled circles: 68 dB SPL. One-sample two-tailed *t*-test. **p<0.01, ns: non-significant, p>0.05. Bar plots show means, and error bars show 95% CI. (**F**) Population-averaged time course of approach behaviour in response to intact (positively weighted) vs temporally irregular (negatively weighted) mouse songs, displayed as in *Figure 2K*. The dark grey trace indicates the mean of normalised, trial-based temporal profiles across all sessions (n = 24). The black bar indicates time bins in which the cumulative approach behaviour significantly deviates from zero (one-sample two-tailed *t*-test). (**G**) Temporal profiles of approach behaviour to intact vs temporally irregular songs over the course of four sound presentation trials (x-axis, coloured bars) for each of the behavioural sessions (y-axis. n=24), displayed as in *Figure 2L*. (**H**) Female approach behaviour during simultaneous playback of intact songs (top) and temporally super-regular songs (bottom), displayed as in panel (**E**). (**I**) Population time course of approach behaviour to intact (positively weighted) vs temporally super-regular (negatively weighted) mouse songs, displayed as in panel (**F**). (**J**) Temporal profiles of approach behaviour to intact vs temporally super-regular songs over the course of four sound presentation trials, displayed as in panel (**G**).

*E*, left, and *Supplementary file 6*). Interestingly, females were highly sensitive to the disruption of song temporal regularity and preferentially approached intact over irregular song playback (mean preference index in response to intact vs irregular songs significantly differed from zero, one-sample *t*-test, t(23) = 3.43, p=0.00023, *Figure 4E*, right). The sensitivity to temporal regularity was robust and similarly strong across all sizes of speaker zones (*Figure 2—figure supplement 2E*). The temporal profile of approach behaviour of a majority of animals favoured intact, regular songs over the course of a session (*Figure 4G*), and across sound presentation trials (*Figure 3—figure supplement 1A*). On average, mouse listeners started to show this sustained preference for regular over irregular songs after 85.4 s of sound playback (normalised cumulative time in speaker zone significantly different from 0, one-sample *t*-test, p<0.05; *Figure 4F*). This longer latency to behavioural discrimination after song

onset compared to when approaching intact song playback over silence (16.9 s, *Figure 2K*) might reflect the need for additional accumulation of evidence and temporal integration when discriminating between the two concurrent sound streams. These striking results in response to disruptions of the song temporal regularity contrast with the lack of noticeable effect of other manipulations of the song and syllable structure on female approach behaviour.

Finally, we asked whether female mice, being obviously attracted by stronger rhythmic regularity in the songs, would preferentially approach an artificial set of 'super-regular' songs over intact ones. To this end, we created a set of highly rhythmic songs in which the silent breaks were modified such that the onset of the syllables in each song occurred exactly at the dominant inter-syllable interval, effectively narrowing the distribution of inter-syllable times (interquartile range of syllable interval in super-regular songs = 24 ms compared to 61 ms for intact songs; *Figure 4C, D and H*, left, and *Supplementary file 7*). Female listeners did not show any consistent behavioural preference when the playback of intact songs was contrasted with that of super-regular songs (one-sample $t$-test, $t$(21) = 1.48, p=0.15; *Figure 4H*, right). This was reflected in the time course of approach behaviour, which showed large variability across animals over a session with no clear emerging pattern (*Figure 4J*). All four sound presentation trials elicited comparable response profiles (*Figure 3—figure supplement 1C*). This behavioural invariance to improvements of the songs' temporal regularity remained when tested at different timepoints during song playback (*Figure 4I*) or using different spatial parameters for analysis (*Figure 2—figure supplement 2H*). Together, these results suggest that the rhythmicity of natural male songs is already sufficiently salient to reach the criterion for approach behaviour by female listeners and cannot be improved by additional regularisation of syllable onsets.

## Discussion

In this study, we developed an ethological assay to probe the behavioural relevance of several candidate acoustic features in mouse communication. By exploiting female mice's natural approach response to male-produced courtship songs and using competing playbacks of intact and manipulated songs, we directly tested how listeners use acoustic information from vocal sequences. The results show that female mice monitor temporal regularity in male songs, while their behaviour was unaffected by changes in the song sequential organisation or the fine acoustic structure of individual syllables. The findings highlight the selective extraction of song temporal regularity, which may be an acoustic signature of singer fitness, as an effective, potentially evolutionarily conserved behavioural strategy for the sensory compression of complex vocal sequences during goal-directed social behaviour.

On the one hand, neurons across the mouse auditory system are tuned to different spectro-temporal features in song syllables, and mouse listeners have been shown to be perceptually sensitive to, or able to detect, the acoustic manipulations of individual syllables used in the current study. The upper limit of this perceptual ability has been determined using go/no-go operant conditioning to extensively train animals to discriminate between syllables from spectro-temporally distinct categories, between intact and temporally reversed or pure tone versions (*Neilans et al., 2014*), or between partial from whole syllables (*Holfoth et al., 2014*). However, such externally rewarded reinforcement learning and the associated sharpening of sensory acuity and neuronal tuning likely bias and amplify the perceptual ability of a naive animal (*Fritz et al., 2005*; *Fritz et al., 2003*). On the other hand, social behaviour in naturalistic settings requires the selective monitoring of specific informative sensory features, while cognitively suppressing or ignoring other dimensions that may however be perceptible, as classically demonstrated in the toad's visual system during prey capture (*Wachowitz and Ewert, 1996*). Our work complements previous studies that quantified the upper bounds of mouse hearing sensitivity. We show that, in the context of an ethologically valid behaviour, female listeners show behavioural invariance to several acoustic features that they are, or can be trained to become, perceptually sensitive to.

We found that females' approach to male song playback was not consistently affected by changes to the global structure of the vocal sequence, such as the temporal reversal of the song and the randomisation of the syllable order in the song. Both of these manipulations preserve the relative spectro-temporal relationships within each syllable, as well as the physical complexity and acoustic characteristics of the sequence, and tested whether the order of the syllables was informative to the listener. Indeed, songs from male mice have been shown to demonstrate some characteristic sequential features, with short syllables typically dominating the beginning of a vocal phrase (*Chabout*

*et al., 2015*; *Matsumoto and Okanoya, 2016*; *Castellucci et al., 2018*; *Castellucci et al., 2016*; *Hertz et al., 2020*). This structure would be violated both by the syllable order randomisation and the temporal reversal of the song. While mouse listeners are known to perceptually discriminate between syllable types differing in their spectro-temporal features (*Neilans et al., 2014*), our finding that female approach behaviour was not affected by a shuffling of the syllable sequence indicates that, during courtship behaviour, females monitor the presence/absence or relative occurrence of specific syllable types, or other acoustic features of the songs, independent of the syllable order. Our result also disambiguates previous findings on mouse sensitivity to syntactic information in male songs: *Chabout et al., 2015* previously showed, in a similar behavioural paradigm, that female listeners discriminated between syntactically simple and complex songs produced by males in different social contexts. Songs produced in the two tested social contexts differed both in the first-order syllable sequencing and in the composition of the syllable repertoire. Our finding on females' behavioural insensitivity to syllable order thus suggests that listeners may have been primarily exploiting differences in repertoire composition to discriminate and ultimately select songs produced from a specific social context. Thus, our results show that female listeners do not strongly rely on the global structure of the syllable sequence during courtship behaviour.

When manipulating the songs at the more local level of individual syllables, we found that female listeners approached sequences of noise bursts or pure tones at the syllable peak frequency similarly to intact male songs. Our finding on female mice's behavioural invariance to replacing syllables by pure tone sequences replicates previous work that showed female mice approaching playbacks of synthetic 70 kHz ultrasounds over silence, when these were presented from behind one of two devocalised males (*Pomerantz et al., 1983*). In contrast, *Hammerschmidt et al., 2009* used a place preference paradigm comparable to the one used in this study, and an artificial sequence of irregularly timed short ultrasounds that matched neither the duration or syllable rate of mouse songs. In this case, the authors showed that female mice preferentially approached male songs over pure tones. Our results resolve the apparent discrepancy between these previous studies: by showing both females' behavioural invariance to pure tone approximations of syllables and their high sensitivity to temporal regularity, our work suggests that the preference for intact over ultrasound sequence observed in the Hammerschmidt study is driven by the co-occurring disruption in sequence temporal regularity. Thus, the evidence suggests that sound energy in and around a critical band corresponding to the syllable frequency range, displayed at the natural temporal organisation of male songs, is a sufficient approximation of male courtship songs to drive female approach behaviour. Our and others' results on female courtship behaviour complement the body of work on maternal behaviour by lactating mothers: as long as the temporal structure of pup calls is maintained, pup retrieval behaviour is robust to the removal of most of the spectral structure of ultrasonic pup isolation calls (*Ehret and Haack, 1982*), and three-harmonic stack approximations of sonic wriggling calls elicit normal maternal responses (*Gaub and Ehret, 2005*). Similar importance of temporal patterns for the perception of social communication, over the fine acoustic structure of individual elements, has been demonstrated behaviourally in other animal species, including humans: as illustrated in vocoded speech, the intelligibility of speech degraded in the spectral domain remains partially preserved if it is shaped with the correct amplitude envelopes (*Shannon et al., 1995*; *Van Tasell et al., 1987*).

Of the subset of acoustic features tested in this study, temporal regularity in the song was the only feature we observed to be extracted by female listeners from male acoustic courtship displays. Our other tested contrasts show that several other perceptually dramatic disruptions of song features were nevertheless not impacting on the female approach behaviour as long as the temporal patterning of the songs was preserved, suggesting a high degree of saliency for temporal regularity as a socially informative cue. We found that mouse listeners preferentially approached intact over irregular courtship songs, suggesting that regularity is attractive to females in the context of vocal-based mate selection. The temporal regularity observed in mouse songs is a consequence of breathing patterns regulating the production of syllables by the emitter as individual calls are generated following the onset of exhalation (*Tschida et al., 2019*; *Castellucci et al., 2018*; *Sirotin et al., 2014*). Disruption to the temporal regularity of male song, as artificially introduced in this study, may, for instance, result from irregular breathing cycles and thus 'stuttering' singing by the male, or by the inability to sustain bouts of vocal production of sufficient duration to elicit a salient percept of regularity in the listener. A recent study of courtship songs by Foxp2 knockout mice provided direct evidence for a link between

male emitter fitness and song temporal regularity by showing songs by mutant mice contain rhythmic irregularities (*Castellucci et al., 2016*). From the perspective of the listener, temporal predictability has been shown to facilitate auditory detection of near-threshold stimuli compared to an identical aperiodic sequence (*Lawrance et al., 2014*). Additionally, in human listeners, speech intelligibility is disrupted by temporal jittering the sentences (*Ghitza and Greenberg, 2009*; *Pichora-Fuller et al., 2007*). Thus, perhaps by improving the perceptual signal-to-noise ratio of male courtship songs, temporal regularity may represent an acoustic cue signalling the fitness or suitability of the potential mating partner to the listener (*Egnor and Seagraves, 2016*), to which we show that socially motivated listeners are highly attuned to.

Female behaviour did not appear to discriminate between intact songs and an artificial 'super-regular' version of the songs, suggesting that natural mouse songs might already be at ceiling in terms of the perceptual saliency of their temporal regularity, and therefore, their attractiveness to female listeners. In the somatosensory system, neuronal responses in barrel cortex responded indistinguishably to lightly temporally jittered and perfectly regular sequences of whisker deflections, when the stimulus presentation rate was slower than 20 Hz (*Lak et al., 2008*). In a hypothesis that remains to be directly tested in the auditory system, it is thus possible that, at the natural syllable rates of mouse songs, removing the low temporal jitter present in intact songs to create super-regular songs does not significantly impact on perceptually relevant neuronal substrates.

The syllable rate in mouse songs (~7 Hz; *Holy and Guo, 2005*) broadly matches with that of other mammalian vocal patterns such as monkey vocalisations and human speech (*Ghazanfar and Takahashi, 2014*; *Chandrasekaran et al., 2009*). On the one hand, this stimulus presentation rate in mouse songs is directly determined by the breathing rhythm of the singer. Given that active sensation during whisking and sniffing operates within the frequency range of both respiratory-coupled oscillations (*Tort et al., 2018*) and the hippocampal theta rhythm in rodents (*Kleinfeld et al., 2006*; *Kepecs et al., 2007*; *Grion et al., 2016*; *Kleinfeld et al., 2016*), similar constraints may be placed on the cortical mechanisms of hearing (*Schroeder et al., 2010*), especially during natural behaviour. This intriguing hypothesis could be addressed in future work by simultaneously tracking the listener's hippocampal theta oscillations and sniffing behaviour during song playback. On the other hand, the syllable rate in mouse songs generally corresponds to the timescale of slow amplitude envelope in human speech, which is known to contribute to the identification of both prosodic and syllabic content (*Peelle and Davis, 2012*). While vocal sequences such as human speech and mouse songs are quasi-periodic rather than metronomically regular signals, they both contain sufficiently salient temporal regularity to allow listeners to make predictions about the incoming signal (*Peelle and Davis, 2012*). In particular, the comparable range of temporal regularities found in mouse and human vocal sequences corresponds to the theta frequency of cortical oscillations that have been shown to be susceptible to entrainment by streams of regularly presented stimuli (*Peelle and Davis, 2012*; *Doelling et al., 2014*). An influential model of speech processing proposes that cortical theta oscillations in the 4–8 Hz range entrain to the slow temporal envelope modulations in speech and may play a role in parsing continuous sensory input by coordinating neuronal excitability to optimally support the decoding of phonemes (*Giraud and Poeppel, 2012*). Whether similar neuronal dynamics can be observed in response to courtship song sequences in the mouse auditory system, and what neuronal substrates causally support the behaviourally relevant extraction and representation of acoustic features identified in this study, remains to be determined.

## Conclusion

In summary, we used an ethologically driven approach to evaluate the behavioural relevance of several acoustic features in mouse communication. Our results identify a key role of temporal regularity, but invariance to global and local song structure, in the goal-directed use of vocal patterns by female listeners. The findings highlight the selective monitoring of temporal regularity in communication sounds as an effective, potentially evolutionarily conserved behavioural strategy for the sensory compression of complex vocal sequences during mammalian vocal perception.

## Methods

### Animals

A total of 83 female C57Bl/6J inbred mice (*Mus musculus*, Charles River Laboratories, UK) participated in the experiments. All experimental procedures were carried out in accordance with a UK Home Office Project License approved under the United Kingdom Animals (Scientific Procedures) Act of 1986, and in compliance with international legislation governing the maintenance and use of laboratory animals in scientific experiments (European Communities Council Directive of 24 November 1986, 86/609/EEC). Animals were housed in same-sex groups of 3–5 per cage and under a reversed 12 hr light/dark cycle. Food pellets and water were provided ad libitum. Animals were tested in eight batches of 8–12 animals between September 2017 and September 2019. Mice participated in the playback experiments between the ages of 5–11 wk (mean = 9 wk; *Figure 2—figure supplement 1*). In order to prevent any impact of handling stress from influencing behavioural testing, the animals were regularly handled and habituated to all aspects of the behavioural protocol for at least a week before starting experiments. Tube handling (*Gouveia and Hurst, 2013*) was used exclusively. The mice had no sexual experience, but were exposed to two different groups of males through a mesh division for 5 min each following each behavioural testing session. This allowed olfactory, auditory, and visual contact with males while preventing physical contact and intercourse, and has been suggested to help maintain female's motivation for approach behaviour (*Shepard and Liu, 2011*).

### Acoustic stimuli

Ultrasonic vocalisations were obtained from five male C57BL/6 mice in response to urine samples from conspecific females in oestrus, across 11 recording sessions. The urine stimulus consisted of the tip of one sterile cotton bud soaked in a mixture of freshly collected urine from at least two animals. Sounds were recorded at a sampling frequency of 250 kHz (Avisoft Bioacoustics, Germany) using a condenser ultrasonic microphone (CM16/CMPA), recording interface (UltrasoundGate 416H), and software (Avisoft Recorder version 5.2.09, all from Avisoft Bioacoustics).

The stimulus set used in the playback experiments ('intact/original songs') consisted of a subset of seven songs produced by three C57BL/6 male mice (median age 22 wk) over six recording sessions. The sound files were high-pass filtered above 40 kHz and were denoised using the frequency-domain noise reduction algorithm in Avisoft SASlab Pro (version 5.2.09, Avisoft Bioacoustics). Syllable detection and segmentation was verified manually. Sound files were up-sampled to 260,420 Hz with anti-aliasing for playback.

### Acoustic manipulations

An artificial set of randomised syllable sequences was created by shuffling the sequential order of the syllables within each song. Each syllable remained paired with its subsequent silent interval in order to preserve the distribution of inter-syllable intervals across the songs.

The set of temporally reversed songs was creating by playing each original song backwards, for example, flipped along the time axis such that the last sample of the last syllable was played first, and so on, until the first sample of the first syllable which was played last.

Phase-scrambled songs were created by replacing the phase of the original signal with a random phase value for each original song separately. The resulting high-frequency noise syllables were then ramped with a 0.5 ms linear rise/fall time.

Pure tone sequences were created by replacing each syllable with a pure tone of the matching duration, including a 0.5 ms linear rise/fall time. Tone frequency was selected to match the median peak frequency of all recorded syllables, for example, 76 kHz.

For the phase-scrambled and pure tone sequence sets, sound amplitude was constant across all syllables in one song, and was adjusted for each song separately in order to match the mean root mean square amplitude of syllables in the corresponding original songs.

A set of temporally irregular songs was created by replacing the silent pauses associated with each syllable (time intervals between the offset of one syllable and the onset of following syllable) with randomly generated break durations under the condition that the duration of the new song be equal to that of the original song.

A super-rhythmic version of each of the original songs was created by scaling the silent pauses between each pair of successive syllables such that every inter-syllable interval (time between the

onset of one syllable and the offset of the following syllable) matched the median inter-syllable interval of a given song. For syllables that were longer than the median inter-syllable interval, the silent pause was adjusted to 'skip a cycle' such that the next syllable occurred at twice the median syllable interval. The total duration of each super-rhythmic song was matched to that of the corresponding original song by modifying the longer inter-bout interval (silent pause between regular sequences of syllables).

Sounds were played using MATLAB (MATLAB version R20015a; MathWorks, Natwick, MA) and a digital signal processor (RX6, Tucker-Davis Technologies, FL). Acoustic stimuli were delivered via two free-field electrostatic speakers driven (ES1, Tucker-Davis Technologies) placed at ear level, 3 cm from the mesh window at the edge of the enclosure. The frequency response of the loudspeaker was ±6 dB across the frequency range used for stimulation [68–84 kHz]. Sounds were played either at 53–59 dbSPL, or 63–70 dbSPL, as measured just inside the testing box (7 cm from the speakers). Before the start of the first behavioural session of the day, correct playback of ultrasonic stimuli was confirmed visually using the Avisoft-RECORDER software and condenser ultrasonic microphones (Avisoft CM16/CMPA, both by Avisoft Bioacoustics) placed at the edge of the behavioural box.

## Oestrous staging

Oestrous stage was assessed daily based on vaginal cytology (*Byers et al., 2012*; *Caligioni, 2009*). Samples were collected by flushing warmed sterile saline over the vaginal opening. The unstained cell samples collected via vaginal lavage were then visualised using bright-field microscopy (Leica DFC365FX). Oestrous stage was estimated visually based on the relative proportions of leukocyte, nucleated and cornified epithelial cells, following the oestrous cycle stage identification tool outlined in *Byers et al., 2012*. In a subset of animals tested between October 2017 and March 2018, oestrous stage was identified based on visual examination of the vagina (*Byers et al., 2012*; *Caligioni, 2009*). Both oestrous staging methods required brief (10 s) tail restraint and took place daily at similar times in the morning at least 2–3 hr before a behavioural experiment in order to prevent any handling stress from impacting the behaviour.

## Behavioural testing

All behavioural tests were conducted during the dark (active) phase of the light cycle, between 12:00 and 20:00. Animals participated in the experiments once per oestrous cycle, when identified as being in oestrus or proestrus. A minimum of 5 d separated successive test sessions for each mouse. Animals were tested at most once in each given sound contrast. The order of experimental contrasts each animal participated in was assigned pseudo-randomly in order to balance the ages of animals taking part in each type of behavioural tests (no difference in mean listener age across all tested contrasts, one-way ANOVA, $F_{(6, 153)} = 0.013$, p=1.0; *Figure 2—figure supplement 1*).

Behavioural assays took place in a two-compartment Plexiglas behavioural box (22.7 × 25.2 × 22.5 cm) under infrared LED illumination (no visible light, *Figure 2B*). A soundproof partition partly divided the behavioural box down the middle into two 'speaker zones' joined by a larger 'neutral zone'. Wall panels with a mesh insert were positioned at the end of each speaker zone, such that the mesh-covered openings were level with two speakers placed outside of the box, on either side of the partition. This allowed undistorted delivery of acoustic stimuli separately into each of the two speaker zones, while playbacks from both speakers were equally audible in the centre of the neutral zone. The behavioural box was placed inside a double-walled soundproof booth (IAC Acoustics), whose interior was covered by 4-cm-thick acoustic absorption foam (E-foam, UK). Two trays containing 2 g of a mixture of soiled bedding freshly collected from two cages of males were placed below each speaker, outside of the behavioural box. After each behavioural testing session, the wall panels were washed with soapy water and dried, and the behavioural boxed wiped down first with 70% ethanol, then with distilled water.

One behavioural session lasted for 40 min and contained a 10 min silent habituation period and four sound presentation trials (160 s of song playback) each followed by a 3 min break (*Figure 2B*). The session started with the mouse being placed in the middle of the neutral zone using the handling cylinder. After the 10 min silent habituation period, the first playback trial was initiated manually as soon as the animal returned to the middle of the neutral zone. Each playback trial consisted of one concatenated presentation of the songs in the stimulus set (seven songs, 2 min 40 s total sound

duration), played in random order. Each trial (playback of one full stimulus set) was followed by a 3 min silent break, before the next trial could be initiated.

In a 'song vs silence' contrast experiment, the song set was played back through one of the two speakers, while the other remained silent. In a 'original vs manipulated song' contrast experiment, the original song set and corresponding manipulated version of the song set were played back simultaneously, each through a different speaker. The playback onsets of individual pairs of original and manipulated songs were jittered with respect to each other by a time delay randomly sampled between 20 and 90 ms, with the leading stimulus set randomly assigned for each trial. The speaker/playback side for each sound presentation programmes was randomly assigned at the beginning of one session and alternated across the four consecutive trials within each session. This strategy allowed to distinguish, for each session, between a behavioural preference for one of the sound types despite changes in the location of the sound source, and a preference for one of the physical locations (side bias).

## Video tracking

The position of the animal was recorded under infrared illumination using an overhead camera (640 × 480 pixel resolution, PlayStation 3 Eye, Sony) that had been customised by manual removal of its infrared blocking filter. A custom image processing pipeline in Bonsai (*Lopes et al., 2015*) detected the position of the animal against the white background and saved the x and y coordinates of the centroid of the body, as well as the corresponding timestamps (ca. 30 frames/s). On- and offset timestamps for song playback were recorded by manual key presses.

## Data analysis

Tracking data were analysed in MATLAB using custom scripts (version R2018a; MathWorks). For timestamps during which the detection of the mouse's body failed, the x–y position was interpolated by repeating the previously available position. For each playback side, a 'speaker zone' was defined as the area of the behavioural box adjacent to each mesh opening in the wall panel, ranging 86 mm into each arm of the divided section of the box. The relative strength of the animal's place preference for one stimulus over the other was captured using a preference index defined as follows: $Preference index = \frac{t(song) - t(manipulated song)}{t(song) + t(manipulated song)}$ , where *t(song)* is the time in seconds the animal spent in the speaker zone adjacent to the side of original song playback for this trial and *t(manipulated song)* is the time spent in the other speaker zone, for example, the side of manipulated song playback or silence. The preference index was computed for each sound presentation trial. The preference index for one behavioural session was calculated as the median of the four trial-based sound preference indices for that session.

Side bias across a testing session was assessed by quantifying the time in seconds an animal spent in the left and right speaker zone during each of the four sound presentation trials. Sessions in which the animal displayed a clear side bias (consistent preference for the left or right side of the behavioural box despite changes in the sound playback side, evaluated as all four trials showing a positive, or all negative, difference between the times in the left and right speaker zones) were excluded from the analysis (29% ± 7% of behavioural sessions excluded due to side bias, mean and 95% CI).

To evaluate the temporal evolution of each session's approach behaviour, one positive (resp. negative) increment was assigned to each video frame during which the animal was inside the area (speaker zone) in front of the speaker playing back intact (resp. manipulated) songs. Frames during which the animal was outside either speaker zones were assigned a value of zero. Temporal profiles quantifying the accumulation of dwell time in either speaker zones were computed by taking the cumulative sum of these traces along the relevant trial(s) and dividing by the video frame rate to obtain times in seconds. Before comparing or pooling data across animals, each trace from an individual session or trial was normalised to the absolute value of its maximum amplitude. When displaying all traces individually, sessions were sorted by the amplitude of their last element.

All statistical analyses were performed using MATLAB (version R2018A). Data were tested for deviation from normality using Lilliefors' composite goodness-of-fit test (MATLAB function lillietest) and Shapiro–Wilk's parametric hypothesis test of composite normality (MATLAB function swtest) at p<0.05 before using parametric tests. Significant population place preference for each contrast was tested using one-sample two-tailed *t*-test against zero. Unless otherwise stated, all statistical tests

are two-tailed. In all figures, significance levels are indicated as follows: *significant at the 0.05 level, **significant at the 0.01 level.

## Acknowledgements

We thank Margaux Silvestre for help with testing initial experimental designs, Camille Suess for contributing to data collection, and Jennifer Linden for feedback and support. We thank Joshua Jones-Macopson and Erich Jarvis for advice on optimising murine singing behaviour, and developing an approach behaviour paradigm. Tania Barkat, Florian Studer, and Sebastian Reinartz provided helpful comments on the manuscript. This work was supported by the Swiss National Science Foundation (P2SKP3_158691, CP) the Wellcome Trust (110238/Z/15/Z, CP), the Medical Research Council (MR/M002889/1, DB), and a Royal Society Research Grant (RG130622, DB).

## Additional information

### Funding

| Funder | Grant reference number | Author |
|---|---|---|
| Schweizerischer Nationalfonds zur Förderung der Wissenschaftlichen Forschung | P2SKP3_158691 | Catherine Perrodin |
| Wellcome Trust | 10.35802/110238 | Catherine Perrodin |
| Medical Research Council | MR/M002889/1 | Daniel Bendor |
| Royal Society | RG130622 | Daniel Bendor |

The funders had no role in study design, data collection and interpretation, or the decision to submit the work for publication. For the purpose of Open Access, the authors have applied a CC BY public copyright license to any Author Accepted Manuscript version arising from this submission.

### Author contributions

Catherine Perrodin, Conceptualization, Resources, Data curation, Software, Formal analysis, Funding acquisition, Investigation, Visualization, Methodology, Writing - original draft, Writing - review and editing; Colombine Verzat, Resources, Software, Investigation; Daniel Bendor, Resources, Funding acquisition, Methodology, Writing - review and editing

### Author ORCIDs

Daniel Bendor http://orcid.org/0000-0001-6621-793X

### Ethics

All experimental procedures and post-operative care were approved and carried out in accordance with the UK Home Office, subject to the restrictions and provisions contained within the Animal Scientific Procedures Act of 1986. Experiments were conducted under PPL P61EA6A72 (Bendor).

Reviewer #1 (Public Review): https://doi.org/10.7554/eLife.86464.2.sa1
Reviewer #2 (Public Review): https://doi.org/10.7554/eLife.86464.2.sa2
Reviewer #3 (Public Review): https://doi.org/10.7554/eLife.86464.2.sa3
Author Response: https://doi.org/10.7554/eLife.86464.2.sa4

## Additional files

### Supplementary files
- MDAR checklist
- Supplementary file 1. Example of intact song – from top panel in *Figure 3A*.

- Supplementary file 2. Example of randomised song – from bottom panel in *Figure 3A*.
- Supplementary file 3. Example of reversed song – from bottom panel in *Figure 3C*.
- Supplementary file 4. Example of phase-scrambled song – from bottom panel in *Figure 3E*.
- Supplementary file 5. Example of pure tone syllables – from bottom panel in *Figure 3G*.
- Supplementary file 6. Example of irregular song – from bottom panel in *Figure 4E*.
- Supplementary file 7. Example of super-regular song – from bottom panel in *Figure 4H*.

## Data availability

All data used to produce the figures in this study are available in figshare with the identifier (https://doi.org/10.6084/m9.figshare.24474760). The computer code used in this study is available at (https://github.com/bendor-lab/Perrodin-et-al-eLife/, copy archived at *Bendor, 2023*).

The following dataset was generated:

| Author(s) | Year | Dataset title | Dataset URL | Database and Identifier |
|---|---|---|---|---|
| Perrodin C, Bendor D | 2023 | Approach behaviour for female mice listening to male vocalisations (natural or experimentally manipulated) | https://doi.org/10.6084/m9.figshare.24474760 | figshare, 10.6084/m9.figshare.24474760 |

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
