## [Editor Report · eLife assessment]

This **valuable** work advances our understanding of the acoustic features driving the attraction of female mice to male vocalisations. The evidence supporting the conclusions is **solid**, with well-designed place preference assays and manipulations of male song structure. The work will be of broad interest to neurobiologists and ethologists working on mouse social interactions, auditory processing and communication.

---

## [Referee Report · Reviewer #1 (Public Review)]

This work deals with courtship behaviour in mice. Authors try to identify the acoustic features that influence the attractivity level of male courtship songs to females. Courtship songs are made of sequences of short ultrasound syllables emitted at a rate of 7-10Hz. Authors manipulated these syllables by changing either the spectrotemporal content of each syllable or the intersyllable intervals. The authors found that it was only when sequences of syllables were irregular (with highly variable intersyllable intervals) that the female was less attracted to the song. The data, therefore, brings evidence that the acoustic features of syllables account less than the song's temporal regularity for the attractivity of courtship songs. The authors suggest that temporal regularity of syllable emission, building on breathing patterns, could reflect male fitness. They also suggest that temporal regularity could be an acoustic cue compressing the complex acoustic information carried by songs.

Strengths:

The study is well-written, very straightforward, and easy to follow. Behavioral tasks are well-designed and many tests, on a large enough set of animals have been done to support the conclusions. Results are clearly presented and provide enough details to see individual points. The discussion makes interesting connections between syllable rhythms and animals' fitness or brain rhythms.

Weaknesses:

Although the study is easy to understand and provides interesting results, the data analysis remains incomplete, and the interpretation of results is not cautious enough.

For instance, Fig. 2 shows a preference for song playback but we cannot determine if it is a general preference for a sound or a specific preference for male songs because only the difference between the presence of song or silence is tested. I acknowledge that the authors did not overstate their results, but the experimental design is incomplete and hard to interpret in that respect. For instance, the expression "preferential approach to song" is ambiguous.

There is no analysis of individual preference across tests and we might have the feeling that the effect shown mostly depends on the preference of only a few animals. Indeed, it seems that roughly one-third of animals showed a strong preference for the intact song while another third showed a strong preference for the modified song, whatever the modification. A few animals are therefore "swing voters". It would have been interesting, if not pertinent, to have a deeper analysis of the behavior of these later animals. Do they choose less (i.e. spend less time close to speakers) or do they swing from one corner to another? What about the animals which always chose the modified song? Are these animals that already showed a weak or strong preference for silence, therefore showing they were not comfortable with the songs played? There is no discussion of these aspects either.

Also, on page 11, it is written "female listeners perceptually compress the high sensory dimensionality of male songs by selectively monitoring a reduced subset of meaningful acoustic features in isolation." This statement or hypothesis is questionable. After all, if someone would change the inter-syllable intervals in human speech, that would become cryptic or at least annoying for the listener. Humans would definitely prefer normal speech. Is this because we compress acoustic features? Not really. It is likely that this modified speech just differs too much from the set of parameters typically encountered and therefore understood/interpreted while learning a language in childhood. Thus, the hypothesis here is rather to determine, for a given acoustic feature, if there is a range within which the perception of the message carried by the song (courtship) is maintained. Interpretation of "compressed acoustic features" with regards to animals' preference seems an overinterpretation. Same remark at the end of the conclusion.

---

## [Referee Report · Reviewer #2 (Public Review)]

In the present manuscript, Perrodin et al. investigated which properties of ultrasonic vocalizations determine their attractiveness for female mice. They collected a set of male courtship vocalizations and compared their attractiveness for female mice against a number of conditions, including silence, and a number of modified sequences.

The study has a clear design and used insightful modifications on the vocalization sequences, which allow the present results to be linked to previous results. The most interesting outcome of the study is that female mice prefer regularly timed sequences of vocalizations over less regularly timed sequences. This result is novel and adds to our understanding of the determinants of social communication between mice. Overall the study is likely underpowered, which was, however, hard to assess as animal numbers were largely not reported for the individual tests, and statistical analysis was carried out on the level of sessions only.

The study has a very good discussion embedding the current results with the previous literature, although the discussion steps beyond the results in a few respects, in particular when trying to determine the underlying reasons for the preference for regularly spaced sequences.

Methodologically the study is carried out at the appropriate level, although some improvements could be made to the experimental apparatus to avoid reflections.

The study will likely have a substantial impact on the field of mouse communication because the regularity of spacing has not been a focus of previous research. In addition, the confirmation that a lot of other modifications are less determining for the attractiveness of the vocalizations provides solid data on which to base future work.

---

## [Referee Report · Reviewer #3 (Public Review)]

Perrodin, Verzat and Bendor describe the response of female mice to the playback of male mouse ultrasonic songs. The experiments were performed in a Y-maze-like apparatus with two acoustically separate response chambers. Sounds were presented in 4 trials, alternating strictly between the left and right branches of the Y. Cumulative dwell time in the two chambers was measured, and used as an index of female preference. They first show, consistent with previous observations, that female mice will spend more time near a speaker playing a male mouse song than near a speaker playing nothing. They then performed several manipulations-time reversals, syllable order randomization, phase scrambled replacement, pure tone replacement, and 'hyper-regular' inter-syllable-intervals-which female mice did not discriminate from the normal song in this assay. Finally, they show that females spent more time near normal songs than near songs with more variable inter-syllable-intervals

The authors' approach to the problem was ethologically sensible -- females were tested in proestrus and estrus, the male odor was used to increase motivation, mouse handling was with tube transfers to reduce stress, mice were age-matched across conditions, and experiments were conducted in the dark (active) phase. In addition, animals were habituated to handling and to the apparatus.

The acoustics were very good. The acoustic structure of the vocal signals was well described. Specific ranges of dB SPL were reported, speaker flatness was evaluated, the sound amplitude was matched in manipulated and unmanipulated songs, and playback onset timing jittered randomly between manipulated and unmanipulated signals.

I think it is a reasonable result. My concerns are the following:

1. The authors use "approach" as it has been used in other publications, but what is actually measured is dwell time. Pomerantz et al, 1983 observed that female mice approached mute and singing males the same number of times (e.g. approached both at the same rate), but spent more time with the singing than the mute male. Their use of "approach" to describe dwell time was a bit confusing to me, but sticking with the way the literature is defensible. However, they also refer to the assay as a "place preference assay", which I found confusing.

2. I am a bit worried about their method of removing side bias (29% of trials). It certainly seems like a reasonable thing to exclude mice that simply picked one side or the other, but, because the stimulus always alternated between the sides, this exclusion of mice exhibiting a side bias is also excluding, specifically, behavior that would be incorrect.

3. Given the observation by Hammerschmidt et al, 2009, that female mice would only discriminate male songs in a playback assay on the first presentation, it is important to know whether females were used across the different manipulations. How many conditions did each female experience? How often did a female display positive discrimination in a condition after having displayed no discrimination?

Specific comments:

1. For Figure 2L

The heat map legend is labeled "Towards" indicating a motion towards either the speaker playing the song or the silent speaker. However, there is nothing in the methods that indicates that the direction of movement was ever measured. I may have missed it, but I can't figure out how this heat map was generated and what it represents. The figure legend states: "Normalized temporal profiles of approach behaviour to mouse songs vs silence over the course of 4 sound presentation trials (x-axis, coloured bars) for each of the behavioural sessions (y-axis, each animal is one line, n = 29), calculated as in I. Sessions (lines) are ordered by the amplitude of their last element." 2I states " I. Temporal profile of approach behaviour over the four sound presentation trials in the example session in C, calculated as the cumulative sum of time in the intact song playback (positively weighted) vs silent (negatively weighted) speaker zone." I interpret this to mean that "Towards" is an inaccurate description of what is being plotted, as there is no motion, only dwell time.

**References**

K. Hammerschmidt, K. Radyushkin, H. Ehrenreich & J. Fischer (2009) Female mice respond to male ultrasonic 'songs' with approach behavior. Biol. Lett. 5:589-592.

Pomerantz, S.M., Nunez, A.A. & Bean, J (1983) Female behavior is affected by male ultrasonic vocalizations in house mouse. Physiol. Behav. 31:91-96.

---

## [Author Response]

We would like to thank the editor and the three reviewers for their time and effort taken in reviewing our manuscript and providing constructive feedback. Unfortunately, the first author of this manuscript is no longer involved in academia, and does not wish to further revise this manuscript. However, we agree with the entirety of the feedback and critiques provided by the referees, and feel these points should be taken into account when interpreting our results and conclusions.